# Neurogenesis during Brittle Star Arm Regeneration Is Characterised by a Conserved Set of Key Developmental Genes

**DOI:** 10.3390/biology11091360

**Published:** 2022-09-16

**Authors:** Anna Czarkwiani, Jack Taylor, Paola Oliveri

**Affiliations:** 1Department of Genetics, Evolution and Environment, University College London, Darwin Building, Gower Street, London WC1E 6BT, UK; 2Center for Regenerative Therapies Dresden (CRTD), Technische Universität (TU) Dresden, 01307 Dresden, Germany; 3Center for Life’s Origins and Evolution, University College London, London WC1E 6BT, UK

**Keywords:** ophiuroid, echinoderm, regeneration, nervous system, pax6, elav

## Abstract

**Simple Summary:**

Injuries to the central nervous system most often lead to irreversible damage in humans. Brittle stars are marine animals related to sea stars and sea urchins, and are one of our closest evolutionary relatives among invertebrates. Extraordinarily, they can perfectly regenerate their nerves even after completely severing the nerve cord after arm amputation. Understanding what genes and cellular mechanisms are used for this natural repair process in the brittle star might lead to new insights to guide strategies for therapeutics to improve outcomes for central nervous system injuries in humans.

**Abstract:**

Neural regeneration is very limited in humans but extremely efficient in echinoderms. The brittle star *Amphiura filiformis* can regenerate both components of its central nervous system as well as the peripheral system, and understanding the molecular mechanisms underlying this ability is key for evolutionary comparisons not only within the echinoderm group, but also wider within deuterostomes. Here we characterise the neural regeneration of this brittle star using a combination of immunohistochemistry, in situ hybridization and Nanostring nCounter to determine the spatial and temporal expression of evolutionary conserved neural genes. We find that key genes crucial for the embryonic development of the nervous system in sea urchins and other animals are also expressed in the regenerating nervous system of the adult brittle star in a hierarchic and spatio-temporally restricted manner.

## 1. Introduction

Regeneration of the nervous system is a question that fascinated scientists for decades, especially considering that axons of the central nervous system (CNS) notoriously fail to regenerate in humans compared to peripheral nerves. This is due to a variety of limiting factors such as the lack of attractive or trophic factors in the adult brain and spinal cord, or in fact the presence of inhibitory factors of neurite growth in the CNS [1]. In contrast, various other invertebrate and vertebrate species show high regenerative capacity from whole brain regeneration in planarian worms after amputating the animals in half [2], to regeneration of the severed spinal cord [3] or parts of the brain in zebrafish and salamanders [4,5,6]. However, many of the instances of adult neurogenesis observed in these various animal taxa are seen to use an array of different regenerative mechanisms and cellular processes, including stem and progenitor cell proliferation, dedifferentiation of existing cells and even transdifferentiation of cells to replace lost neuronal subtypes [7,8,9,10]. The evolutionary origin of adult neurogenesis is thus not clear and open questions remain concerning the conservation of the regenerative mechanisms displayed by these different animal models and the ability to generate new neurons in homeostatic conditions in mammals. Therefore, understanding the different mechanisms of regeneration and neurogenesis in a variety of animals would improve the potential to find molecular or cellular targets for the development of regenerative therapies for repair of injuries to the CNS.

Brittle stars are members of the echinoderm phylum of marine invertebrates characterised by their pentaradial body plan organisation. Echinoderms belong to the ambulacrarians, which together form a phylogenetic sister group with chordates, and remarkably can regenerate most adult structures including their central nervous system. Both the generation of new neurons under normal physiological conditions as well as neurogenesis during post-traumatic regeneration has been extensively studied in related echinoderms—sea cucumbers [11,12,13,14,15], sea urchins [16,17] and sea stars [18,19,20,21], but to a lesser degree in brittle stars. The brittle star has however been shown to be an excellent model to study the cellular and molecular mechanisms of arm regeneration and embryonic development allowing, therefore a direct comparison of the two processes in the same species. The brittle star body is formed by a central disc that contains the visceral mass and five segmented arms. Each arm segment is a complex structure composed of different organs and tissue types including the radial nerve cord, intervertebral muscles, podia, spines, skeletal shields and vertebrae, ligaments and a radial water canal, all of which can regenerate perfectly after autotomy or amputation to generate again a completely functional arm. Several studies focused on the regeneration of the skeletal elements in the brittle star *Amphiura filiformis*, which showed that the regeneration of the adult skeleton re-uses parts of the established embryonic skeletogenesis gene regulatory network [22,23,24]. Proliferative cells, likely from the coelomic epithelium, differentiate first into migratory skeletal precursor cells and then mature cells capable of depositing the biomineralised skeleton [25,26,27]. Brittle star embryonic neural development has been described in the species *Amphipholis kochii* [28] and *Amphiura filiformis* [29] revealing the extent of the larval and juvenile nervous system. Additionally, recent work has described in detail the anatomy and expression of cell-type specific markers in the nervous system of the adult arm in two brittle star species—*Amphipholis kochii* and *Ophioderma brevispinum* [30,31]. Remarkably, these studies revealed a high degree of conservation of the neuroanatomical architecture of the peripheral nerves and the central radial nerve cord, as well as the heterogeneity of ophiuroid radial glial cells between two species which diversified over 250 million years ago [32,33].

Here, we elucidate the potential molecular drivers of neuronal growth and repatterning during arm regeneration in the brittle star by looking at the expression of key transcription factors regulating neurogenesis across the animal kingdom. Group B SRY-related HMG box genes (*soxB* genes) have been shown to first regulate the formation of the embryonic neurectoderm and also to maintain neurectodermal cells in a progenitor state in a variety of animals from *Nematostella*, through sea urchins to vertebrates [16,34,35,36]. Following the formation of neuronal progenitors, members of the superfamily of basic helix-loop-helix (bHLH) transciption factors such as *atonal* and *neuroD* are involved in the specification of these cells and differentiation into subsets of neurons [37,38,39]. This has also as recently been neatly demonstrated in the developing octopus brain and although the sequence of expression of bHLH genes, as well as the extent of differentiation of the progenitors in which they are expressed varies slightly across species, they all direct neural progenitor commitment [40]. After the stage of neural fate acquisition, regulatory genes such as *musashi*, and *embryonic lethal abnormal vision* (*elav*) are involved in the final differentiation stages of neurogenesis [16,17,41]. Elav protein expression has already been shown to be expressed in virtually all neurons of the radial nerve cord in the intact brittle star arm [30]. In sea urchins specifically, *six3* has been shown to be required for the development of all neurons in the embryo [16,42]. Finally, *pax6* expression has not been extensively studied in the sea urchin embryo but it is expressed in adult sea urchin tube feet where it is presumed to have a role in photoreception [43], and thus is also an indicator of the final differentiation and maturation of the sensory nervous system. We thus intend to expand our understanding of neural regeneration in brittle stars by studying the expression of genes required for driving the specification and differentiation of neuronal subtypes as well as the patterning of the intricate nervous system at different regeneration stages of the arms of the brittle star *Amphiura filiformis*.

## 2. Methods

### 2.1. Animal Maintenance and Collection

*Amphiura filiformis* species were obtained in the fjord close to the Kristineberg Center for Marine Research and Innovation, Sweden, at depths of 20–60 m using a Petersen mud grab. Before any experiments, animals were always left to acclimatise for a few days in flow-through tanks (14 °C) in London in filtered artificial seawater (Instant Ocean, Acquarium Systems; 30‰ salinity). During all manipulations live *Amphiura* were anaesthetised in a solution of 3.5% MgCl_2_·6H_2_O in a 50:50 solution of filtered sea water (FSW) to milliQ H_2_O. For 50% differentiation index (DI) and 95% DI regenerated stages arms were amputated at the intersegmental boundary at a fixed distance from the disc dependent on its size in accordance with standardised tables composed previously [44] or at 1 cm from the disc for stages 1–5 [26]. A maximum of 2 arms out of 5 were cut per brittle star to minimize stress placed on the animal. The animals were then returned to sea water and left to regenerate for a given amount of time to reach the desired differentiation index (blastema, 50%, 95%) [44] or early regeneration stage [26]. Following this step, the arms were collected either for whole mount in situ hybridisation experiments or for RNA extraction. For the former, the regenerates were cut along with a few pre- regenerative segments retained as a control. For RNA extraction, 30 regenerates per stage were used and the amputation was limited to the tissue of interest only (1 mature segment for non-regenerating and stage 1 samples, regenerated tissue only for stages 2–5, several segments of proximal or distal tissue excluding distal cap structure) as described. Tissue for WMISH were then fixed in 4% PFA overnight at 4 °C and washed with 1X PBST. For long term storage the regenerates were put in 100% methanol at −20 °C. Samples for RNA were placed in RLT (Qiagen) and stored at −80 °C until extraction.

### 2.2. Immunohistochemistry

Regenerating arm samples that were fixed for WMISH as above in 4% PFA were used for the majority of antibody experiments with the exception of the samples for anti-synaptotagmin B staining, which were fixed in PFA for 15 min at room temperature and postfixed in 100% methanol for 1 min. Arms were first rehydrated using a descending Ethanol: H_2_O gradient (70%, 50%, 30%) at room temperature (RT). Following 3 PBST (phosphate buffered saline +0.1% Tween-20) washes (RT), a permeabilization step involving PBSTX (PBS with a 1:100 Triton-X100) was used for 1hr at RT before washing with a blocking buffer (PBST + 4% Goat serum) for 30 mins at RT. The primary antibodies used in this work are listed in Appendix A. The samples were incubated in primary antibody solution overnight at 4 °C, washed in PBST and incubated in secondary antibody solution (1:1000) for 2 h at room temperature (Appendix A). DAPI was used to label nuclei. All arms were oriented and imaged from the oral side, where the regenerating RNC is best visible, unless otherwise specified. Imaging was carried out using a Zeiss AxioImager A1 microscope, Zeiss light-sheet Z1 (Carl Zeiss, Jena, Germany) or Leica TCS SP2 confocal microscope (Leica, Wetzlar, Germany).

### 2.3. Whole-Mount In Situ Hybridisation

WMISH was carried out according to the protocol outlined previously [24], with no modifications. All arms were oriented and imaged from the oral side, where the regenerating RNC is best visible, unless otherwise specified. Imaging of WMISH samples was carried out using a Zeiss AxioImager A1 microscope. A list of probes used in this work is provided in Appendix A

### 2.4. NanoString—nCounter Analysis

The NanoString nCounter dataset used in this study was created, validated and described in detail in a previous publication [24]. Briefly, total RNA was extracted, and cDNA was synthesised from a pool of regenerating arms as described before [24,25]. The samples were processed in the Nanostring facility at UCL using 100 ng of total RNA/sample and with the nCounter (NanoString Technologies, Seattle, WA, USA) according to manufacturer’s instructions. The results were analysed using the nSolver software 4.0 (NanoString) and quantified as described previously [24].

## 3. Results

The anatomy and composition of the brittle star arm has been described in detail in the species *Amphipholis kochii* and *Ophioderma brevispinum*, suggesting that the architecture of the brittle star arm central and peripheral nerves is conserved within ophiuroids [30]. Together with this study, previous work describing the arm anatomy specifically in *Amphiura filiformis* [26,27,45] confirm that the overall organisation of the arm nervous system in *A. filiformis* is similar to that in other brittle stars. The central nervous system composed mainly of the radial nerve cord is positioned in the oral part of the segments and extends throughout the arm (Figure 1A). Interestingly, after amputation there is a high degree of proliferation observed specifically at the oral position within the segments corresponding to the position of the radial nerve cord [26]. To characterise the organisation of the nervous system in regenerating arms of *A. filiformis* following amputation we stained for antibodies marking different neuronal subpopulations in the different regions of a 50% stage regenerating arm, which showcases all the levels of segment maturation. Anti-synaptotagmin B (SynB) staining highlights not only the radial nerve cord running through the middle oral side of the proximal differentiating segments of the regenerating arm (Figure 1B), but also the peripheral nerves strongly expressing this protein in rings at the base of podia (Figure 1D), which then extend projections along the podium into the tip (Figure 1D’), similar to what has been observed in the mature segment of *O. brevispinum* [30]. Interestingly, there is no SynB signal observed in the newly-formed distal undifferentiated segments (Figure 1C), except for the distal tip, which contains some mature synaptotagmin-expressing cells (Figure 1C’). This observation is in line with the proposed regeneration mechanism suggested for the brittle star, whereby a differentiated terminal structure is first formed and then intercalation of newly emerging segments between the stump and the distal tip takes place to rebuild the lost arm [26]. Acetylated-tubulin immunoreactivity is primarily observed in putative ciliated cells in the outermost structures of the arm including podia, spines and the growing arm segments (Appendix A), but strong labelling can also be seen in proximal muscle nerves (Appendix A, arrows) similarly to the case in *O. brevispinum*. Gamma-tubulin is detectable in neural projections extending within both podia and spines (Appendix A). Serotonergic neurons can be found in localised tracts along the aboral side of the arm segments and additionally a few single serotonin-positive cells are localised to the tips of the spines. Large clusters of serotonin-positive cells are also found at the tips of the podia (Appendix A).

We then proceeded to determine the expression patterns of key regulatory genes underlying neurogenesis during early and late stages of brittle star arm regeneration. During early stages of regeneration (stage 2–3) of *A. filiformis*, *Afi-elav, Afi-soxC, Afi-soxb1, Afi-six3, Afi-pax6* (Figure 2; early stages panels) and *Afi-neuroD* (Appendix A; early stages panels) have a strong expression in the distal region of the regenerate. *Afi-elav, Afi-soxB1* and *Afi-neuroD* then become localised to only a few cells at the tip (stage 4–5), while *Afi-pax6* and *Afi-soxC* are strongly expressed in the developing distal structure. Interestingly, their expression precedes the morphological differentiation of the distal structures including the terminal podium surrounded by the terminal ossicle [26]. *Afi-atonal* and *Afi-nkx2.1* have no detectable expression at this stage (Appendix A). In addition to strong distal expression, *Afi-six3* is also expressed in a line of cells corresponding to the regenerating radial nerve cord and *Afi-pax6* is also expressed in podia primordia. *Afi-soxB2* is only localised to the epidermis at first, where sensory neurons will appear, then once the arms reach stage 4 it begins to be expressed in a highly specific repetitive pattern in a more lateral position relative to the expression of *Afi-six3* in the regenerating radial nerve cord. It appears that this pattern precedes the morphological differentiation of the newly forming metameric units (Figure 2; early stages panel). *Afi-musashi*, a gene encoding for a RNA-binding protein highly expressed in CNS neural progenitor cells [46], is expressed in the epidermis and distal tip of stage 2–3 arms, and then has strong expression in the distal end of stage 5 arms throughout the radial nerve cord as well as patterned expression in podia primordia in the proximal end of the stage 5 arm (Appendix A). At late stages of regeneration *Afi-elav Afi-nkx2.1*, *Afi-pax6* and *Afi-musashi* are all expressed in the distalmost structure, although *Afi-pax6* and *Afi-musashi* show expression only in the terminal podium (Figure 2 and Appendix A; 50% distal panel). The remaining genes are not expressed in this structure but rather mark distinct territories in the newly forming and differentiating neuronal domains. *Afi-elav*, *Afi-soxB1*, *Afi-soxC*, *Afi-six3*, *Afi-atonal*, *Afi-neuroD*, *Afi-musashi* and *Afi-nkx2.1* share a highly localised segmental pattern of expression in the oral side of the regenerating arm corresponding to the cell bodies of the radial nerve cord, which are localised in the centre of individual segments and do not span inter-segment regions compared to their axons (Figure 2; 50% proximal panel; [30])*. Afi-soxB2* is expressed in the epidermis of the distal region and then together with *Afi-pax6* and *Afi-musashi*, is expressed in the regenerating podia although in very different patterns—*Afi-musashi* is localised to the tip of the podium, *Afi-soxB2* is expressed in scattered single cells in the podia and *Afi-pax6* is expressed all along the length of the podia, corresponding to the nerve plexus. *Afi-elav*, *Afi-soxB1, Afi-soxB2, Afi-soxC* and *Afi-pax6* are also expressed in the podia of adult non-regenerating brittle star arms though in distinct patterns (Figure 2; podium panel). *Afi-elav, Afi-soxC* and *Afi-pax6* are expressed in a ring around the base of the tip of the podium, while *Afi-soxB1* is expressed at the very distal tip of the podium. The expression patterns of these neuronal genes reveal the positions of different elements of the nervous system such as the epidermis, podia and the terminal podium, which likely give rise to different subtypes of peripheral sensory neurons, and the cells in the radial nerve cord.

Next, we wanted to study in depth the quantitative changes in expression of the various neuronal genes throughout regeneration using a high-throughput gene expression analysis system—the nanostring nCounter. We used probes from a previously published codeset [24] and looked at the emerging patterns of gene expression at early and late stages of arm regeneration (stage 1–24 h post amputation, 48 hpa, and 72 hpa; blastema stages 3, 4 and 5 and late regeneration stages from 50% differentiated arms’ proximal and distal segments [26]). Sample collection was carried out as outlined in Figure 3A; note that for the distal segments of the 50% regenerated arms we excluded the distal cap structure, which shows signs of differentiation [26].

Many of the neuronal genes are strongly downregulated 24 h post amputation relative to the non-injured, non-regenerating arm, namely *Afi-elav*, *Afi-soxB2*, *Afi-soxB1*, *Afi-musashi*, *Afi-neuroD*, and *Afi-six3* (Figure 3B). Additionally, we observed many Delta/Notch signaling pathway components, such as *Afi-delta*, *Afi-notch*, and *Afi-serrate* as well as the FGF signaling gene previously shown to be expressed in the regenerating brittle star radial nerve cord *Afi-fgfr1* [24], being downregulated at this timepoint as well (Figure 3B). After 48 hpa a few genes are beginning to become upregulated, such as the FGF pathway ligand *Afi-fgf9/16/20* and neuronal *genes Afi-soxB2, Afi-soxB1* and *Afi-soxC* (Figure 3B). At the blastemal stages additional upregulation of *Afi-musashi*, *Afi-brn1/2/3* and *Afi-six3* can also be detected together with *Afi-fgfr1* and *Afi-serrate* (Figure 3B). Consistently, *Afi-nkx2.1*, which could not be detected in the early stages using in situ hybridisation (Appendix A) also shows no expression using the nanostring until late stages of regeneration (Figure 3B). In the 50% proximal and distal segments, most of the neuronal genes show the highest levels of expression. *Afi-elav* and *Afi-pax6* are only strongly expressed at this stage compared to earlier stages (Figure 3B). It is worth noting that two well-known developmental genes important for the specification of an embryonic neural territory called the apical organ, namely *Afi-foxJ1* and *Afi-foxQ2* are not expressed in adult brittle star arms. Interestingly, when we looked at the expression of proliferation-associated and stem-cell associated genes such as cyclins (*Afi-cycA*, *Afi-cycB* and *Afi-cycE*), the transcription factors *Afi-pcna*, *Afi-mycb* and *Afi-piwi*, they are also strongly downregulated right after amputation and begin to be turned on between 48 and 72 hpa, and are very strongly upregulated in blastemal stages and late regeneration stages (Figure 3C). This is consistent with EdU-labeling, which shows a shutdown of proliferation at 24 hpa and a progressively higher number of cells in S-phase starting at 72 hpa and throughout the arm regeneration in *Amphiura filiformis* [26]. We thus observe two main patterns of the dynamics of gene expression during regeneration as shown in Figure 3D. Proliferation and stem-cell associated genes are generally not highly expressed in non-regenerating arms, then start to be switched on around 48 hpa, and are highly expressed in blastema stages 3–4 as well as in undifferentiated distal segments of late stage regenerating arms. There is a lower level of expression of these genes in the proximal segments of 50% regenerated arms, which are undergoing differentiation (Figure 3D). On the other hand, we see genes with the opposite expression pattern, which generally are expressed at lower levels throughout regeneration compared to non-injured arms with the exception of the proximal segments of the 50% stage regenerates (Figure 3D). These opposing patterns of expression are suggestive of the potential role of these genes in the onset of regeneration, such as proliferation or cell fate specification (early stage patterns in red) or alternatively in the differentiation, patterning or maintenance of cell types at late stages and in fully differentiated non-regenerating arms (late stage patterns in blue). Altogether, these data reveal the strength of looking at the gene expression patterns at strict developmental stages in trying to identify potential functions that the genes may have during the process of arm regeneration in the brittle star.

## 4. Discussion

The ability to regenerate a complete central nervous system is extraordinary among deuterostomes, and so understanding how it occurs in brittle stars may yield important information to focus efforts for regenerative medicine. We show here that the brittle star *Amphiura filiformis* can completely restore not only the complete central radial nerve cord and its projections but also peripheral nerves which provide a sensory function for the podia. This regeneration is characterised by a timely expression of known developmental genes, which have been shown to be important for neuronal ectoderm specification, differentiation and maturation of the array of mature neuronal subtypes during development in various species. Although unlike in the context of development, the timing of expression of these genes during regeneration does not follow the same straightforward order. Supporting previous hypotheses concerning the distalisation-intercalation mode of the brittle star arm regeneration [26], neurogenic gene expression patterns seem to confirm the formation of a distal structure very early on during regeneration (stage 2/3), which although not yet prominent histologically, is already defined by a co-expression of not only the early genes which specify the neuroectodermal territory such as *Afi-six3* and *Afi-soxB1*, but also genes known to be important for the final differentiation stage of neurons such as *Afi-musashi* and *Afi-elav* (summarised in Figure 4). At stage 4/5 this terminal differentiated structure is even more obvious with the co-expression in this specific territory of the majority of genes studied including *Afi-pax6*, which most likely correlated to the formation of the terminal podium. Nevertheless, the region just below this terminal structure both at the early stages and late stages of regeneration (50% distal) is free from neurogenic gene expression until clear segments and/or podia appear. It seems then that there is a growth zone just proximal to the terminal tip that adds new metameric units in which the new radial nerve cord and its axonal extensions begin to differentiate and mature, reminiscent of the recently proposed mechanism of indeterminate homeostatic growth for adult brittle star arms [31]. This appears to be supported by the opposing temporal gene expression of putative stem cell and proliferation-related genes such as *Afi-mycb, Afi-piwi* and *Afi-vasa* compared to the neuronal genes *Afi-elav*, *Afi-neuroD* and *Afi-pax6* (Figure 3). Together these data suggest that although a clear terminal structure is formed at the distal-most end, the region just underneath it is likely to be the growth zone where a high degree of cell proliferation [26] is responsible for the regeneration of the radial nerve cord. It is however not clear if it arises from a pool of progenitor cells or whether the extending nerve cord from the non-amputated part of the arm (seen histologically and with the expression of *Afi-six3*) can dedifferentiate to give rise to new neurons. It would also be very interesting to understand whether the early regeneration of the nerve cord drives the formation of the whole arm, such as the nerve dependent regeneration of many animals including salamander limbs [47,48] and even starfish [49]. *Afi-soxB2* expression suggests its role in the laying out of the future metameric units as well as the formation of the peripheral neuroectodermal territories such as the ciliated epidermis and podia. This is in line with the role of *soxB2* in posterior growth in annelids [50] and ciliogenesis and patterning of the sea urchin embryo [51]. Perhaps the nerve is a driving force for patterning the new metameric units and could even be a source for signalling through pathways known to be present in brittle star regenerating arms such as FGF [24] and Delta/Notch [52].

## 5. Conclusions

We provided a first glimpse into the anatomy and regulatory genes involved in brittle star neuronal regeneration, which will provide a platform for understanding adult neurogenesis in deuterostomes using an experimental model which so readily regrows and repatterns components of both the central and peripheral nervous system.

## Figures and Tables

**Figure 1 biology-11-01360-f001:**
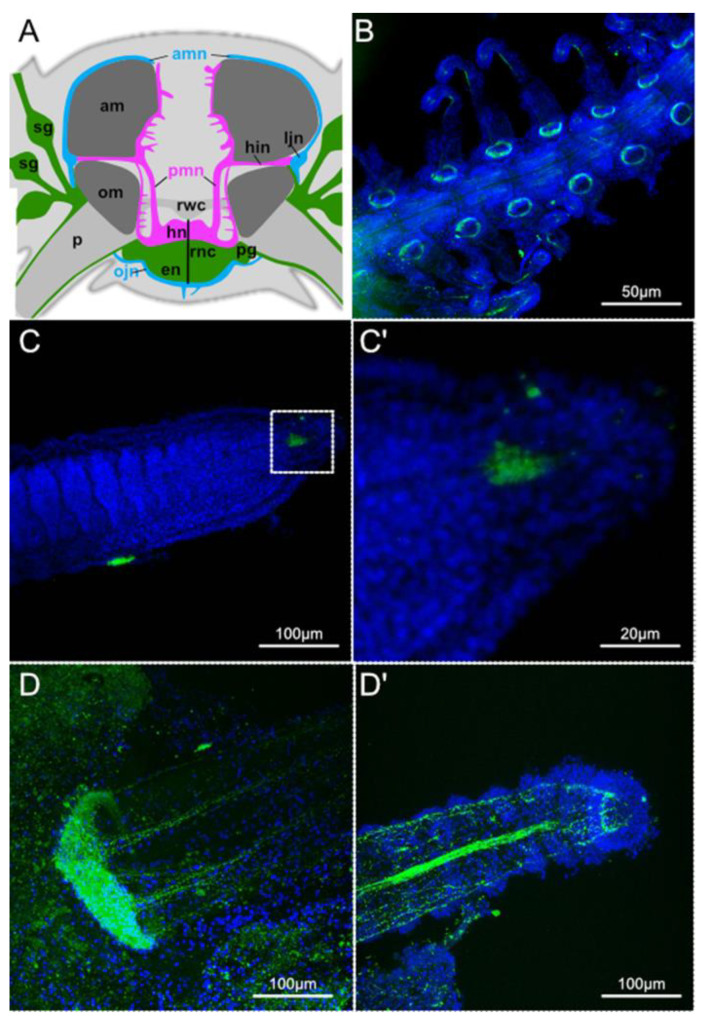
Nervous system in the brittle star arm. (**A**) Schematic diagram of the distribution of the nervous system in a mature arm segment of a brittle star, adapted from [30] under the terms of Creative Commons Attribution 4.0 International Licence. (**B**) SynB staining in regen-erating 50% proximal segments of A. filiformis. (**C**) SynB staining in regenerating 50% distal seg-ments of A. filiformis. (**C’**) Magnification of (**C**). (**D**) SynB staining in mature podium base of A. fili-formis. (**D’**) Tip of the same podium from (**D**). am—aboral intervertebral muscle; amn—aboral mixed nerve; en—epineural epithelium, hn—hyponeural epithelium, hin—horizontal intermuscular hyponeural nerve; ljn—lateral juxtaligametal node; ojn—oral juxtaligamental node; om—oral in-tervertebral muscle; p—podium; pg—podial ganglion; pmn—proximal muscle nerve; rnc—radial nerve cord; rwc—radial water-vascular canal; sg—spine ganglion.

**Figure 2 biology-11-01360-f002:**
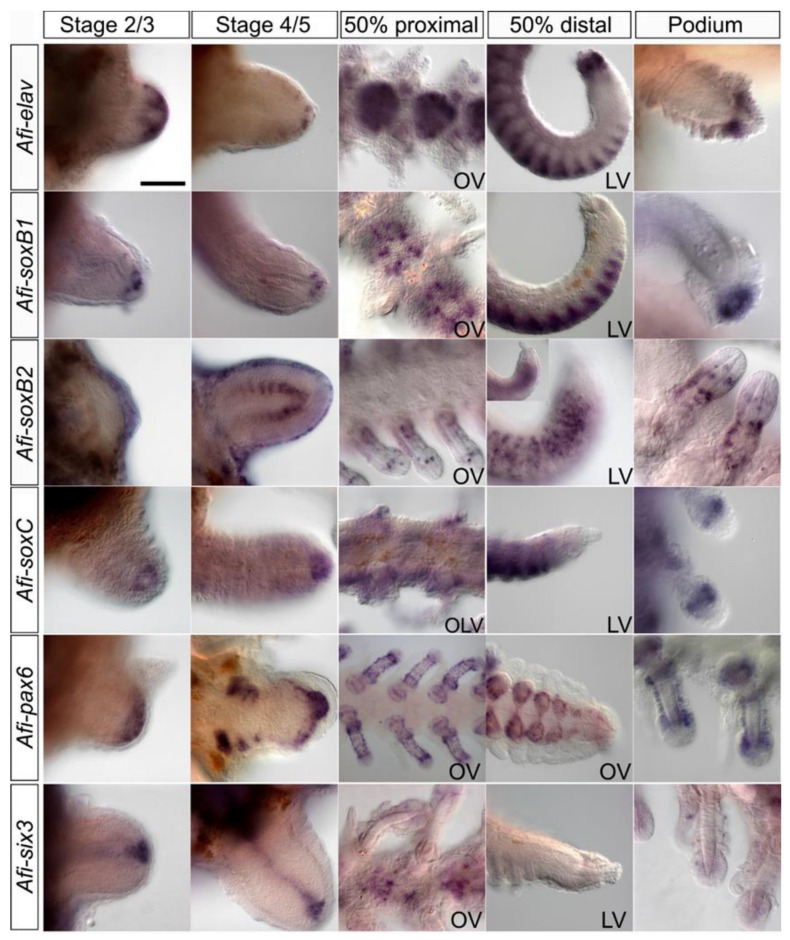
Whole mount in situ hybridisation of *Afi-elav*, *Afi-soxB1*, *Afi-soxB2*, *Afi-soxC*, *Afi-pax6*, and *Afi-six3* in *A. filiformis* at early (stage 2/3 and stage 4/5) and late (50% proximal and distal segments) stages of arm regeneration and podia showing a variety of patterns within the different regions of the regenerating nervous system. OV—oral view, OLV—oral-lateral view, LV—lateral view. Scale bar 100 um.

**Figure 3 biology-11-01360-f003:**
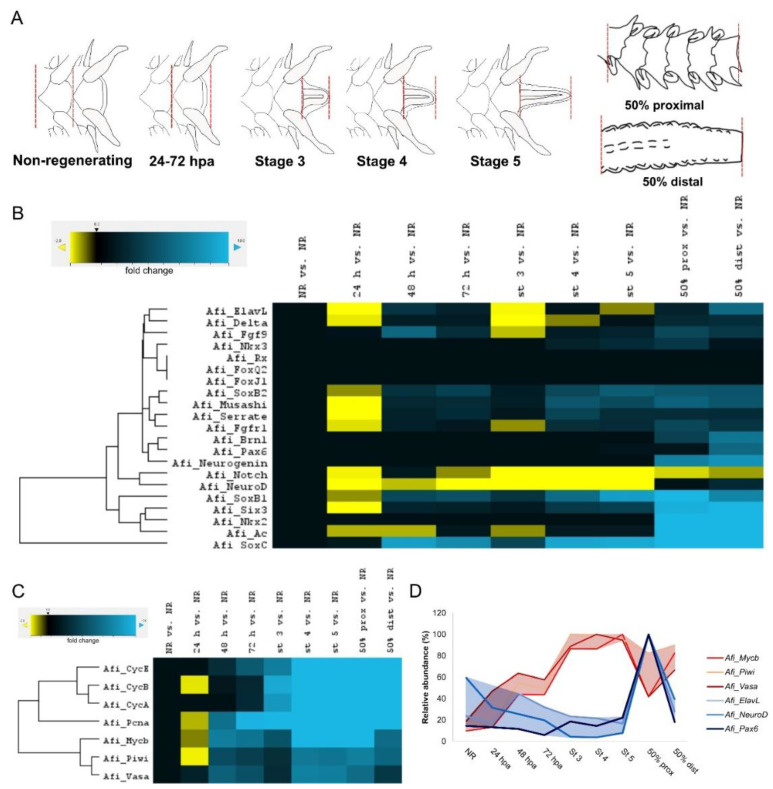
Dynamics of gene expression during arm regeneration of the brittle star *Amphiura filiformis*. (**A**) Schematic diagram representing the tissue collection procedure for generating quantitative expression data using the nanoString nCounter for different stages of arm regeneration. Red dashed lines indicate the tissue being collected at each stage. (**B**) Heatmap and clustering analysis of quantitative data for nervous system gene expression at different stages of regeneration relative to the non-regenerating arm segments obtained using the nSolver package. (**C**) Heatmap of nanoString data for proliferation and stem cell gene expression at different stages of regeneration relative to the non-regenerating arm segments. (**D**) Comparison of the two main observed patterns of expression changes during regeneration using stem cell/proliferation genes *Afi-mycb*, *Afi-piwi* and *Afi-vasa* and nervous system differentiation genes *Afi-elav*, *Afi-neuroD* and *Afi-pax6*. Note the opposite trend in expression suggestive of the different functionality of the genes during regeneration. Expressed as abundance relative to maximum expression (%). NR—non-regenerating, hpa—hours post amputation, St—stage, prox—proximal, dist—distal.

**Figure 4 biology-11-01360-f004:**
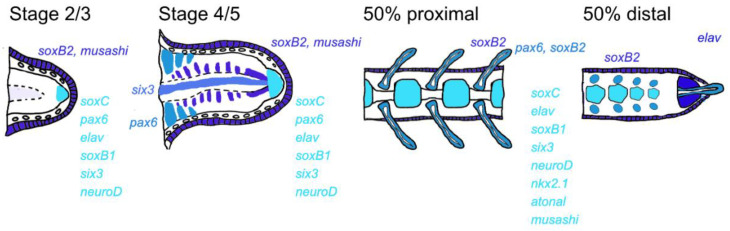
Summary of neurogenic gene expression patterns during early and late regenerative stages of the brittle star *A. filiformis*.

## Data Availability

The datasets of the current study are available from the corresponding author on reasonable request.

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
