# Peer review of "Neurogenesis during Brittle Star Arm Regeneration Is Characterised by a Conserved Set of Key Developmental Genes"

_biology, 2022, doi:10.3390/biology11091360_

Round 1

Reviewer 1 Report

The manuscript by Czarkwiani and colleagues addresses a fundamental problem of body parts regeneration and its molecular mechanisms. Echinoderms represent a powerful model for understanding genetic logic of developmental processes, which now can be tested on diverse taxa. Thanks to the efforts of the author's team, ophiuroids became an attractive organism to study developmental patterning and its evolutionary conservation in deuterostomes. In this context, the manuscript describes very interesting novel data on neural regeneration in anatomical and transcriptional aspects. The authors present spatiotemporal profiles of neural-specific markers, such as conventional immuno-reactivity and gene expression patterns.  This study revealed a unique manner of using conserved neural regulators in the regenerating brittle star arm. The results and conclusions confirm the previously stated hypotheses and significantly clarify our understanding of the restoration of neural structures in echinoderms.

The only flaw I have to mention is the absence of biological replicates in the Nanostring experiment. Either it's not described properly, or it really hasn't been attempted. In the latter case the quantitative data cannot be analyzed statistically, which indicates the insufficient validity of these results.

As a minor remark, I noted that Table S1 has two unfilled rows, and the italics is not always used in Latin and genes names.

Author Response

We thank the reviewers for their time and consideration of our manuscript. Here we address the concerns that they have outlined:

Reviewer #1

The manuscript by Czarkwiani and colleagues addresses a fundamental problem of body parts regeneration and its molecular mechanisms. Echinoderms represent a powerful model for understanding genetic logic of developmental processes, which now can be tested on diverse taxa. Thanks to the efforts of the author's team, ophiuroids became an attractive organism to study developmental patterning and its evolutionary conservation in deuterostomes. In this context, the manuscript describes very interesting novel data on neural regeneration in anatomical and transcriptional aspects. The authors present spatiotemporal profiles of neural-specific markers, such as conventional immuno-reactivity and gene expression patterns.  This study revealed a unique manner of using conserved neural regulators in the regenerating brittle star arm. The results and conclusions confirm the previously stated hypotheses and significantly clarify our understanding of the restoration of neural structures in echinoderms.

The only flaw I have to mention is the absence of biological replicates in the Nanostring experiment. Either it's not described properly, or it really hasn't been attempted. In the latter case the quantitative data cannot be analyzed statistically, which indicates the insufficient validity of these results.

The data from the Nanostring experiment has been collected and described in our previous publication (Czarkwiani et al, 2021), where it is described in detail how the samples were processed and the data validated using biological and technical replicates with two different techniques – QPCR and transciptome quanitification data. For that publication we did not go into detail about the expression patterns of nervous system related genes, which we do describe here. We added a clarification of this in the methods section.

As a minor remark, I noted that Table S1 has two unfilled rows, and the italics is not always used in Latin and genes names.

We adjusted Table S1 accordingly and double checked the italics on the gene names and latin.

Reviewer 2 Report

in the present study, the authors have shown the brittle star arm regeneration with special emphasis on neuronal growth and repatterning. The study is good and the authors have shown the expression of the important genes through histology and RNA expression profiling. However, they have not shown whether these genes are important for regeneration. 

It would be important for the manuscript if they show that some of the genes are important for neuronal regeneration for the proximal and distal parts of the amputated arms by inhibiting the expression of some representative genes and because of this manipulation the regeneration process is inhibited. 

Author Response

We thank the reviewers for their time and consideration of our manuscript. Here we address the concerns that they have outlined:

Reviewer #2

in the present study, the authors have shown the brittle star arm regeneration with special emphasis on neuronal growth and repatterning. The study is good and the authors have shown the expression of the important genes through histology and RNA expression profiling. However, they have not shown whether these genes are important for regeneration. 

It would be important for the manuscript if they show that some of the genes are important for neuronal regeneration for the proximal and distal parts of the amputated arms by inhibiting the expression of some representative genes and because of this manipulation the regeneration process is inhibited. 

We assume that by “inhibiting the expression” the reviewer means gene knockout or knockdown experiments. Unfortunately the brittle star is not amenable to these types of experiments, especially in the adult context. Hopefully this will be possible in the future. We did describe signalling pathway inhibition experiments in our previous publication but these were not within the scope of this manuscript.

Round 2

Reviewer 2 Report

I recommend accepting the manuscript.